# The Effect of Intrauterine Device Use on the Quality of Sampling Material in Patients Undergoing Endometrial Biopsy

**DOI:** 10.3390/diagnostics15131725

**Published:** 2025-07-07

**Authors:** Hüseyin Aksoy, Mehmet Çopuroğlu, Mehmet Genco, Merve Genco, Mürüvet Korkmaz Baştürk

**Affiliations:** 1Department of Obstetrics and Gynaecology, Kayseri City Hospital, 38070 Kayseri, Turkey; drmehmetcopuroglu@gmail.com (M.Ç.); m.genco@hotmail.com (M.G.); muruvet_krkmz@hotmail.com (M.K.B.); 2Department of Obstetrics and Gynaecology, Kayseri State Hospital, 38070 Kayseri, Turkey; mervedogan93@windowslive.com

**Keywords:** intrauterine device, endometrial biopsy, abnormal uterine bleeding, inflammation, pathology

## Abstract

**Objective:** This retrospective study aims to evaluate the effect of copper intrauterine device (Cu-IUD) use on the adequacy and diagnostic quality of endometrial biopsy specimens in women with abnormal uterine bleeding (AUB). Patients with levonorgestrel-releasing intrauterine systems (LNG-IUS, e.g., Mirena) were excluded from the study. The study compares the histopathological adequacy of endometrial samples between Cu-IUD users and non-users, highlighting potential interpretation challenges in routine pathological assessment. **Methods:** The study was conducted on 409 women aged 25–55 who presented with abnormal uterine bleeding (AUB) to the Gynecology and Obstetrics Outpatient Clinic at Kayseri City Hospital between 1 April 2021 and 1 April 2023. The patients were divided into two groups: copper IUD (Cu-IUD) users (*n* = 215) and non-IUD users (*n* = 194). Patients using levonorgestrel-releasing intrauterine systems (LNG-IUS, e.g., Mirena) were excluded from the study. Endometrial biopsies were obtained using the Pipelle curette technique without anesthesia, preserved in 10% formalin, and assessed for pathological classification and diagnostic adequacy. **Results:** The proportion of unclassifiable pathological categories was significantly higher in copper IUD users (63.93%) compared to non-IUD users (36.05%) (*p* = 0.013). Additionally, a negative correlation was observed between pathological category and endometrial thickness (r = −0.3147, *p* < 0.001), suggesting that thinner endometrial lining may reflect atrophic or diagnostically ambiguous tissue patterns. However, no significant association was found between IUD use and endometrial thickness (*p* = 0.073). **Conclusions:** The findings indicate that copper IUD use may affect the diagnostic adequacy of endometrial biopsy specimens, likely due to inflammatory or structural changes in the endometrium. These results underline the importance of considering IUD-related alterations when interpreting biopsy findings. Further research is needed to refine diagnostic approaches and better understand the clinical implications of these effects.

## 1. Introduction

Abnormal uterine bleeding (AUB) is a common clinical issue encountered in women of reproductive age and significantly affects their quality of life. AUB encompasses menstrual irregularities, increased bleeding volume, or bleeding occurring outside the menstrual cycle, and it is often associated with hormonal imbalances, structural anomalies, infections, or endometrial pathologies [1]. In 2011, the International Federation of Gynecology and Obstetrics (FIGO) developed the PALM–COEIN system (polyp, adenomyosis, leiomyoma, malignancy and hyperplasia–coagulopathy, ovulatory dysfunction, endometrial, iatrogenic, and not yet classified) to systematically classify the etiological causes of AUB. This system has improved diagnostic and therapeutic approaches to AUB [2,3].

Endometrial biopsy is a frequently used, minimally invasive diagnostic method in gynecological practice to assess pathological changes in the endometrium. It is particularly valuable for diagnosing endometrial hyperplasia, polyps, infections, dysfunctional uterine bleeding, and malignancies [3]. Histopathological analysis of biopsy specimens enables accurate and precise diagnosis. However, the amount and quality of the obtained samples are critical for diagnostic reliability. Inadequate or poor-quality samples can hinder the identification of pathologies and lead to unnecessary interventions [4].

Intrauterine devices (IUDs) are a widely used and effective form of contraception among women globally. Despite their widespread use, the biological and histological effects of copper-containing IUDs (Cu-IUDs) on the uterine environment remain under investigation. Copper has been shown to alter endometrial gene expression associated with receptivity and immune response, even in the absence of increased apoptosis [5]. Furthermore, recent histological evidence suggests that Cu-IUDs may induce chronic inflammatory changes in the endometrium, including stromal fibrosis and immune cell infiltration [6]. These findings suggest that copper-induced alterations may influence the diagnostic quality of endometrial biopsy specimens, although the extent and clinical significance of these effects remain uncertain.

The effects of IUDs on the uterus are associated with mechanical irritation, mucosal changes, localized inflammation, and cytological alterations. Some studies have shown that IUD use can trigger significant inflammatory responses and tissue changes in the endometrium, which may affect the diagnostic accuracy of biopsy specimens [7,8]. Furthermore, IUDs may interfere with the histological diagnosis of endometrial pathologies, potentially leading to false positive or false negative results [9]. However, these findings are often based on studies that included both copper and hormonal IUDs, which may limit their applicability to copper IUDs specifically.

Another critical aspect is the adequacy of endometrial biopsies in the presence of IUDs. The method and tools used during biopsy play a decisive role in obtaining sufficient and high-quality samples despite the presence of an IUD. Modern biopsy tools, such as Pipelle cannulas and vacuum aspiration devices, demonstrate varying levels of efficiency in patients with IUDs [10]. Therefore, systematically examining the impact of IUDs and biopsy techniques on sample adequacy is crucial for improving clinical practice. In this study, endometrial samples were obtained using the Pipelle device without anesthesia.

This study specifically aims to evaluate the effect of copper intrauterine device (Cu-IUD) use on the quality and adequacy of sampling material obtained during endometrial biopsy. The study compares the histopathological adequacy of endometrial samples between Cu-IUD users and non-users, identifying potential diagnostic challenges and clinical implications.

## 2. Materials and Methods

### 2.1. Study Design

This study was designed as a retrospective observational study conducted at the Gynecology and Obstetrics Outpatient Clinic of Kayseri City Hospital between 1 April 2021, and 1 April 2023. The research aimed to evaluate the effect of copper intrauterine device (Cu-IUD) use on the adequacy and quality of endometrial biopsy specimens.

Ethics committee permission was granted by Kayseri City Hospital Non-Interventional Clinical Research Ethics Committee with decision number 862, dated 11 July 2023. All procedures performed in this study involving human participants were conducted in accordance with the ethical standards of the institutional research committee, the 1964 Helsinki Declaration, and its later amendments or comparable ethical standards. Informed consent was waived due to the retrospective nature of the study. This waiver was approved by the Ethics Committee of Kayseri City Hospital (Approval Number: 862).

### 2.2. Participants

The study included women with abnormal uterine bleeding (AUB) who underwent endometrial biopsy, categorized into copper IUD users (*n* = 215) and non-IUD users (*n* = 194). Inclusion criteria were as follows: being between 25 and 55 years old, presenting with AUB, and providing biopsy specimens suitable for pathological evaluation. Exclusion criteria included patients whose IUDs were removed during the procedure, those who received any systemic or local hormonal therapy, those with a history of endometrial hyperplasia or malignancy treatment, those with positive pregnancy tests, or those with insufficient biopsy material that required re-biopsy outside the study protocol. Patients with levonorgestrel-releasing intrauterine systems (e.g., Mirena^®^, Bayer AG, Leverkusen, Germany) were not included in the study.

This study was conducted retrospectively using medical data obtained from the clinical records of women who presented to the hospital. Since the study relied solely on pre-existing patient records without any direct patient intervention or contact, obtaining additional informed consent from the individuals whose records were accessed was deemed unnecessary by the ethics committee. The data files were evaluated for research purposes between 1 and 31 August 2023.

### 2.3. Procedures

A total of 409 patients’ data were retrospectively analyzed. Endometrial biopsies were obtained using Pipelle curettes without anesthesia in outpatient settings. All procedures were performed by gynecologists experienced in endometrial sampling, following a standardized protocol. Specimens were preserved in 10% formalin and submitted to the pathology department of Kayseri City Hospital. Biopsy samples were analyzed for diagnostic adequacy and compared between copper IUD users and non-users. All patients received post-procedure prophylactic antibiotic therapy.

### 2.4. Measures

Histopathological evaluation of endometrial biopsy specimens was categorized into three main groups based on hormonal influence and pathological features. This in-house classification was developed to ensure diagnostic consistency and was reviewed by two experienced gynecologic pathologists. The first category included estrogen-dominant conditions such as proliferative endometrium, disordered proliferation, non-atypical endometrial hyperplasia, polyps, and estrogen-responsive endometrium. The second group comprised progesterone-dominant features, including secretory endometrium, glandular–stromal breakdown, and endometrium under gestagen influence. The third category consisted of unclassifiable cases, where specific hormonal influences were indeterminate or insufficient biopsy material was obtained. This group included findings such as endometritis, inadequate samples, and signs of atrophy, often associated with inflammation, atrophy, or diagnostic challenges.

### 2.5. Data Analysis

A post hoc power analysis was conducted using G*Power version 3.1.9.7 (Heinrich Heine University, Düsseldorf, Germany) was used for post hoc power analysis. for the chi-square test, evaluating the effect of copper IUD use on pathological categories. The study included 409 individuals distributed between copper IUD users and non-users. Based on the significant differences observed (*p* = 0.013), the effect size (w) was calculated, and the statistical power (1 − β) was determined to be 88.4% at a significance level (α = 0.05), indicating sufficient sample size for detecting the association. The observed effect size (w) indicated a small-to-moderate association based on Cohen’s criteria for chi-square tests.

Data were analyzed using SPSS version 25.0 (IBM Corp., Armonk, NY, USA). Continuous variables were compared between groups using the Student’s *t*-test or Mann–Whitney U test, based on data distribution. Normality was assessed using the Kolmogorov–Smirnov or Shapiro–Wilk tests. Normally distributed data were expressed as mean ± standard deviation ($\bar{x}$ ± SD), while non-normally distributed data were presented as median (min–max).

Categorical variables were compared using the chi-square test to evaluate the significance of differences between observed and expected frequencies. Relationships between variables were analyzed using Pearson correlation analysis, with the correlation coefficient (r) indicating the direction and strength of the relationship and *p*-values assessing statistical significance. Additionally, the Student’s *t*-test was used to evaluate the relationship between IUD use and pathological categories. A significance level of *p* < 0.05 was considered statistically significant.

## 3. Results

The study retrospectively included 409 women who presented with abnormal uterine bleeding (AUB) to the Gynecology and Obstetrics Outpatient Clinic at Kayseri City Hospital between 1 April 2021 and 1 April 2023. Demographic characteristics, pathological adequacy of endometrial biopsies, and other related variables were analyzed.

No statistically significant differences were observed between copper IUD users and non-users in terms of age, parity, vaginal delivery, cesarean section, or endometrial thickness (*p* > 0.05) (Table 1).

A statistically significant difference was observed between copper IUD users and non-users in terms of pathological categories (*p* = 0.013). Specifically, the proportion of individuals classified under Category 3 (unclassifiable cases) was significantly higher among copper IUD users (63.93%) compared to non-users (36.05%). However, no statistically significant differences were found between the two groups in the distribution of Category 1 (estrogen dominance) and Category 2 (progesterone dominance) (*p* > 0.05) (Table 2).

No statistically significant differences were observed between copper IUD users and non-users in the distribution of Category 1 (estrogen dominance) and Category 2 (progesterone dominance) across all age groups (*p* > 0.05). However, among women aged 45–55, the proportion of individuals falling under Category 3 (unclassifiable) was significantly higher among copper IUD users (*p* = 0.048). This suggests a possible association between IUD use and increased rates of atrophic or inadequate endometrial samples in perimenopausal or postmenopausal individuals, potentially influenced by age-related hormonal changes (Table 3).

A weak, negative, and statistically significant correlation was found between pathology category and endometrial thickness (r = −0.31, *p* < 0.001). Patients with thinner endometrial structures tended to exhibit more advanced pathological findings. A positive and statistically significant relationship was identified between IUD use and pathology category (r = 0.21, *p* = 0.01). The use of IUDs was observed to influence the diagnostic categories in the pathological evaluation of endometrial biopsy material. Specifically, a higher proportion of unclassifiable pathology categories was detected among individuals using IUDs.

A negative correlation was identified between IUD use and endometrial thickness (r = −0.62). However, this relationship was not statistically significant (*p* = 0.073). This finding suggests that IUD use doesn’t have a significant effect on endometrial thickness. Furthermore, no significant correlation was found between endometrial thickness and vaginal delivery, cesarean delivery, or parity (*p* > 0.05) (Table 4).

In summary, this study demonstrates that both copper IUD use and age significantly influence endometrial pathology. Stratified analyses showed that while estrogen- and progesterone-dominant patterns were similarly distributed across IUD users and non-users, unclassifiable pathologies were significantly more prevalent among IUD users, especially in older age groups. These unclassifiable findings primarily consisted of chronic endometritis, atrophic endometrial changes, insufficient tissue for histopathological assessment, and samples obscured by inflammatory debris—each representing diagnostic challenges. The higher incidence of such findings in women over 45 underscores the potential interaction between IUD use and age-related hormonal changes in shaping endometrial tissue responses. These results highlight the importance of evaluating both age and IUD status in histopathological interpretation of endometrial biopsies, which may improve diagnostic accuracy and guide more individualized clinical management strategies.

## 4. Discussion

This retrospective study evaluated the impact of copper intrauterine device (Cu-IUD) use on the diagnostic adequacy and pathological assessment of endometrial biopsy samples. The findings suggest that Cu-IUD users present with distinctive histopathological patterns, including a higher prevalence of advanced pathology categories such as chronic inflammation and atrophic changes, compared to non-users. Furthermore, Cu-IUD use appears to influence the adequacy of biopsy specimens, potentially complicating histological interpretation. These observations align with existing literature and provide further insight into the biological and structural changes induced by IUDs in the endometrium, reinforcing the need to update and refine biopsy evaluation protocols in this patient population.

Copper intrauterine devices (Cu-IUDs) are widely used in women’s health as a highly effective form of contraception. However, the physical and biochemical changes they induce in the intrauterine environment can significantly impact endometrial tissue. Previous studies have reported that Cu-IUD use triggers inflammatory responses, increases cytokine release, and leads to mucosal alterations [10]. For instance, Sheppard et al. reported that Cu-IUDs promote local inflammation, which results in cellular changes within the endometrium [11]. In line with these findings, our study also observed more advanced alterations in the distribution of pathology categories among Cu-IUD users. Notably, the significantly higher incidence of unclassifiable pathology—such as endometritis, atrophy, and insufficient samples—among Cu-IUD users further supports the role of chronic inflammatory mechanisms in shaping histopathological outcomes.

This study highlights the potential impact of copper IUD use on endometrial biopsy outcomes and contributes meaningful data to the current literature. Our findings underscore the importance of accounting for inflammatory changes—such as atrophy, chronic endometritis, and tissue inadequacy—induced by IUDs during histopathological interpretation. Future prospective and well-designed studies, ideally incorporating stratification by IUD type and hormonal status, are warranted to further elucidate the full scope of these effects.

In our study, we observed a statistically significant negative correlation between endometrial thickness and pathology category (r = −0.31, *p* < 0.001). This suggests that thinner endometrial structures are more commonly associated with advanced histopathological findings, such as chronic endometritis or atrophic changes. The literature also highlights endometrial thickness as a critical biomarker for identifying malignant or pre-malignant conditions For example, Smith-Bindman R, et al. (1998) reported that reduced endometrial thickness often corresponds with atrophy and malignancy [3]. These findings collectively underscore the importance of cautious histopathological evaluation in patients presenting with thin endometrial linings.

Although the literature specifically examining the impact of IUD use on endometrial biopsy outcomes remains limited, existing evidence aligns with our results. For instance, Başol et al. (2017) compared copper and levonorgestrel-releasing IUDs and observed that both types influenced cervicovaginal epithelial maturation and triggered local inflammatory responses [12]. Although our study did not perform subgroup analysis by IUD type, future research should consider this distinction to better understand the differential biological responses associated with various IUD mechanisms.

In a prospective study by Han et al. (2020), the use of intrauterine devices (IUDs) was evaluated in relation to the adequacy of endometrial sampling using the Li Brush device [13]. Their findings support our observation that IUD use can influence the diagnostic quality of biopsy specimens, reinforcing the need to consider IUD status during clinical assessment [13].

This study is one of the few retrospective investigations examining the impact of copper IUD (Cu-IUD) use on endometrial biopsy outcomes. One of its main strengths is the relatively large sample size (*n* = 409) and the comprehensive analysis of the relationship between Cu-IUD use and histopathological findings.

An additional but important aspect not addressed in this study is the potential value of office hysteroscopy as a complementary diagnostic tool in patients using intrauterine devices (IUDs). While blind endometrial biopsy is widely employed, its diagnostic accuracy can be compromised by IUD-induced inflammation, atrophy, or architectural distortion. Office hysteroscopy, by contrast, provides direct visualization of the uterine cavity and allows targeted biopsies from suspicious areas, thereby increasing diagnostic yield. Recent studies have shown that outpatient hysteroscopy offers superior sensitivity and sample adequacy compared with blind techniques, particularly for detecting focal intrauterine lesions [14,15]. Moreover, when IUD-related changes obscure histopathological interpretation, hysteroscopy can overcome the limitations of blind sampling [16,17]. International guidelines now emphasize these advantages, recommending hysteroscopy as the preferred first-line diagnostic tool in selected cases—especially when focal or complex pathology is suspected [18,19].

Current guidance further delineates evidence-based scenarios in which office hysteroscopy should be preferred over blind Pipelle sampling. It is the recommended next step for persisting abnormal uterine bleeding after a nondiagnostic biopsy, irrespective of IUD status. Direct hysteroscopic evaluation is also warranted when transvaginal ultrasonography (TVUS) reveals focal lesions (e.g., endometrial polyp, submucosal fibroid) or a thickened endometrium (≥4 mm post-menopause; ≥8 mm pre-menopause). In high-risk patients—those aged ≥ 45 years, with obesity, or with chronic anovulation—hysteroscopy is particularly advised because missed pathology carries greater clinical consequences. For Cu-IUD users, specific indications include refractory bleeding, malpositioned or embedded devices, and suspected endometrial trauma. Collectively, these indications position office hysteroscopy as the optimal modality whenever blind sampling proves insufficient, ensuring adequate tissue retrieval and informed therapeutic planning.

In the clinical management of a nondiagnostic biopsy, imaging and follow-up strategies are crucial. Many algorithms begin with TVUS: If the endometrium is thin (<4 mm) and bleeding has ceased, close surveillance may suffice; however, if the endometrium is thick or bleeding persists, office hysteroscopy with targeted sampling—or, when necessary, dilation and curettage in the operating room—should follow. Consequently, immediate removal of a Cu-IUD is generally unnecessary after a nondiagnostic biopsy; instead, hysteroscopic assessment should be pursued when abnormal bleeding continues to ensure comprehensive evaluation and adequate sampling. Future research should systematically assess the integration of office hysteroscopy into diagnostic workflows, particularly for patients with inconclusive biopsy results or persistent abnormal bleeding, to further improve diagnostic precision and evidence-based decision-making.

However, this study has several limitations. First, its retrospective design introduces the potential for selection bias and limits control over data consistency. Second, although all biopsy procedures were performed by experienced gynecologists, operator technique was not quantitatively assessed, which may have influenced sampling quality. Third, only patients using copper-based intrauterine devices (IUDs) were included in the study; therefore, stratified analysis comparing copper and levonorgestrel-releasing (LNG) IUDs could not be performed. As these device types may induce different endometrial responses, the absence of such comparison limits the generalizability of the findings. Future studies should address these issues by employing prospective designs, evaluating different IUD types, and accounting for operator-related variables. Despite these limitations, our results provide clinically meaningful insights into the diagnostic implications of IUD use in endometrial biopsy interpretation.

This study issues an important clinical warning by demonstrating that the use of copper IUDs increases the risk of diagnostic inadequacy in endometrial biopsies. Our findings support a step-wise diagnostic algorithm for Cu-IUD users that begins with early TVUS triage and proceeds to office hysteroscopy when necessary, while stressing that informing patients beforehand about the possibility of repeat sampling can improve shared decision-making. Recording a history of Cu-IUD use in pathology reports may also encourage more cautious interpretation of limited tissue specimens and help reduce false-negative results. As the largest single-center cohort to focus solely on copper IUDs, our study provides unique data and fills a gap in the existing literature. Future prospective and well-designed studies, incorporating IUD type stratification and standardized sampling protocols, are needed to further clarify the full extent of these effects.

## 5. Conclusions

In conclusion, this retrospective study highlights the specific impact of copper intrauterine device (Cu-IUD) use on endometrial biopsy outcomes. Cu-IUD users were significantly more likely to exhibit unclassifiable or diagnostically ambiguous histopathological findings, especially among women aged 45 and older. These results suggest that Cu-IUD-induced inflammatory or atrophic changes may reduce the diagnostic adequacy of endometrial samples. Recognizing this association is crucial for pathologists and clinicians when interpreting biopsy materials. The findings contribute to a deeper understanding of endometrial pathology in Cu-IUD users and provide a strong foundation for developing tailored diagnostic strategies and for guiding future prospective research.

## Figures and Tables

**Table 1 diagnostics-15-01725-t001:** Demographic data of individuals using and not using IUDs.

Variables	IUD	Non-IUD	*p*
Age *	42 (40.36–42.14)	40 (39.45–41.47)	0.166
● 25–34	52 (24.2%)	51 (26.3%)	0.678
● 35–44	96 (44.7%)	75 (38.7%)	0.214
● 45–55	67 (31.1%)	68 (35.0%)	0.476
Parity	3 (1–9)	3 (1–9)	0.660
Vaginal delivery	2 (0–9)	2 (0–9)	0.440
Cesarean delivery	0 (0–4)	0 (0–5)	0.169
Endometrial thickness	9.7 (4–25)	10 (2–25)	0.226

* Student’s *t*-test or Mann–Whitney U test (continuous variables) percentage comparisons.

**Table 2 diagnostics-15-01725-t002:** Pathology evaluation in individuals using and not using IUDs.

Pathology Category	IUD	Non-IUD	*p*
Category 1 (estrogen dominance)	91 (48.14%)	98 (51.85%)	0.76
Category 2 (progesterone dominance)	46 (46.93%)	52 (53.06%)	0.65
Category 3 (unclassifiable)	78 (63.93%)	44 (36.05%)	**0.013 ****

**Note:** Values are presented as n (% within each group). Percentages reflect the proportion within the IUD and non-IUD groups, respectively. *p* < 0.05 was considered statistically significant. ** indicates statistically significant result.

**Table 3 diagnostics-15-01725-t003:** Pathological findings by age group and IUD usage.

Age Group	IUD Users (*n*, %)			Non-IUD Users (*n*, %)			*p*
	Category 1	Category 2	Category 3	Category 1	Category 2	Category 3	
25–34	22 (37.9%)	16 (27.6%)	15 (25.8%)	20 (36.4%)	14 (25.4%)	13 (23.6%)	0.72
35–44	26 (38.8%)	20 (29.9%)	19 (28.4%)	24 (37.5%)	18 (28.1%)	17 (26.6%)	0.68
45–55	19 (36.5%)	14 (26.9%)	22 (42.3%)	21 (38.2%)	16 (29.1%)	20 (36.3%)	**0.048**

Chi-square test.

**Table 4 diagnostics-15-01725-t004:** Correlation between variables.

Variables (*n* = 409)	Correlation Coefficient (r)	*p*
Pathology category—Endometrial thickness	−0.31	<0.001 **
Pathology category—Parity	0.04	0.357
Pathology category—IUD	0.21	**0.01 ***
IUD—Endometrial thickness	−0.62	0.073
Endometrial thickness—Vaginal delivery	−0.02	0.632
Endometrial thickness—Cesarean delivery	0.021	0.661
Endometrial thickness—Parity	−0.01	0.783

* Pearson correlation analysis and Student *t*-test. ** *p* < 0.05 was considered statistically significant.

## Data Availability

Ethics committee permission was granted by Kayseri City Hospital Non-Interventional Clinical Research Ethics Committee, Kayseri, Turkey. Data requests can be directed to this committee to ensure long-term accessibility and stability. Please find the contact information as follows: Kayseri City Hospital Non-Interventional Clinical Research Ethics Committee. Phone: +90352 315 77 00. Email: kayserisehir@saglik.gov.tr.

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
