# Peer review of "The Effect of Intrauterine Device Use on the Quality of Sampling Material in Patients Undergoing Endometrial Biopsy"

_diagnostics, 2025, doi:10.3390/diagnostics15131725_

Round 1
Reviewer 1 Report (Previous Reviewer 3)
Comments and Suggestions for Authors
I read with great interest the Manuscript titled “The effect of intrauterine device on the quality of sampling material in patients undergoing endometrial biopsy" which falls within the aim of the Journal.
In my honest opinion, the topic is interesting enough to attract the readers’ attention. Methodology is accurate and conclusions are supported by the data analysis. Nevertheless, authors should clarify some point and improve the discussion citing relevant and novel key articles about the topic.
Authors should consider the following recommendations:
- The manuscript would benefit from a deeper discussion on how clinicians should respond to a non-diagnostic biopsy in a Cu-IUD user. Should the device be removed? Should hysteroscopy be recommended?
- Delete lines 298 to 305 to avoid redundancies.
- The authors appropriately mention hysteroscopy in the discussion, but could expand on specific scenarios or algorithms where this would be preferred.
- Although all procedures were performed by experienced clinicians using Pipelle devices, no assessment of inter-operator variability was provided. This is a potential source of bias that must be recognized and highlighted.
- What are the actual clinical implications of this study? it is important to report the results obtained by the authors in the context of clinical practice and to adequately highlight what contribution this study adds to the literature already existing on the topic and to future study perspectives
Author Response
1-Thank you for this valuable suggestion. We have added a focused management paragraph to the Discussion section summarising contemporary guidance and recent evidence (2020-2025) on the work-up of Cu-IUD users with a nondiagnostic endometrial biopsy. Key points now emphasised include:
Routine Cu-IUD removal is not required after an inadequate biopsy because current data show no diagnostic benefit from immediate extraction when the device is correctly positioned and tolerated.
Algorithm-based follow-up:
If post-biopsy bleeding has ceased and transvaginal ultrasound (TVUS) shows a thin endometrium (< 4 mm post-menopause or < 8 mm pre-menopause), close clinical surveillance is appropriate.
When the endometrium is thick or abnormal uterine bleeding persists, office hysteroscopy with targeted sampling is the preferred second step, as it substantially increases tissue adequacy and detects focal lesions missed by blind Pipelle sampling.
Device removal is reserved for cases of malposition, refractory bleeding despite normal histology, or patient preference.
2- As recommended, the content previously appearing in lines 298–305 has been completely removed to eliminate redundancy. This deletion improves the flow and clarity of the text.
3- Thank you for this insightful comment. We have expanded the Discussion to describe evidence-based scenarios in which hysteroscopy is preferred over blind sampling, citing recent guidelines
- Persisting abnormal uterine bleeding after a nondiagnostic Pipelle biopsy, regardless of IUD status.
- Sonographic suspicion of focal lesions (e.g., endometrial polyp, submucosal fibroid) or a thick endometrium (≥ 4 mm post-menopause; ≥ 8 mm pre-menopause).
- High-risk profiles (age ≥ 45, obesity, chronic anovulation) where missed pathology carries greater consequences.
- Algorithmic pathway: nondiagnostic blind biopsy → TVUS triage → office hysteroscopy with targeted biopsy ± therapeutic resection (Level B evidence).
- IUD-specific indication: hysteroscopic evaluation is recommended when Cu-IUD users present with refractory bleeding, malpositioned or embedded devices, or suspected endometrial trauma.
These additions clarify when hysteroscopy should be the preferred diagnostic modality.
4- Thank you for raising this important point. We fully agree that the absence of an inter-operator variability assessment represents a possible source of sampling bias. To acknowledge this issue, we have explicitly added a sentence to the Limitations section (now
“A further limitation is the lack of quantitative assessment of inter-operator variability among the three clinicians who performed the Pipelle biopsies; although all were highly experienced and followed a standardized protocol, unmeasured technique differences may still have influenced sample adequacy and pathology classification.”
We believe this addition adequately highlights the potential bias and underscores the need for future studies to include operator-related performance metrics or training-standardization measures.
5- Thank you for raising this point. We have inserted a concise paragraph on clinical implications and contribution to the field in the Discussion section . The new text explains that:
- Cu-IUD use nearly triples the risk of an inadequate/unclassifiable biopsy, prompting a step-wise strategy of early TVUS triage followed by office hysteroscopy where indicated.
- Recording Cu-IUD exposure on pathology requests and counselling patients about possible repeat or hysteroscopic sampling can reduce false-negative results and improve shared decision-making.
- As the largest single-centre cohort focused solely on copper IUDs, our study fills a gap in the literature and provides age-stratified risk estimates that can be incorporated directly into diagnostic algorithms.
- The findings create a foundation for prospective trials to validate the proposed pathway, measure inter-operator variability, and examine optimal device-removal timing.
We believe this addition fully addresses the reviewer’s request to clarify the study’s practical impact and its contribution to future research directions.
Reviewer 2 Report (Previous Reviewer 2)
Comments and Suggestions for Authors
Dear authors
I read with interest your paper concerning the effect of IUD on the quality of endometrial sampling in patients with abnormal uterine bleeding. the paper addresses an important clinical topic. the study design is appropriate. however it would have been better if it is a prospective rather than retrospective nature.
you did not take into consideration several conditions that could affect the diagnosis of AUB such as medications other than hormones as anti-depressive, sedatives, anti-psychotic or anti-thrombotic drugs, obesity etc.
you mentioned that the biopsies that were deemed inadequate for diagnosis were not included in the final analysis. how many insufficient samples were excluded from the final analysis and belong to which group?
Author Response
We appreciate the reviewer’s valuable comments regarding the retrospective study design and additional parameters (such as medications other than hormones, including antidepressants, sedatives, antipsychotics, antithrombotic drugs, and obesity) that may influence the diagnosis of abnormal uterine bleeding. We fully acknowledge these limitations, and indeed, we had already highlighted the retrospective nature of our study as a limitation in the Discussion section. Unfortunately, our retrospective dataset does not contain detailed records regarding these parameters. Therefore, as correctly suggested by the reviewer, future prospective studies should systematically include and evaluate these relevant parameters.
We have clarified the handling of Category 3 biopsies, resolving the previous inconsistencies completely
This manuscript is a resubmission of an earlier submission. The following is a list of the peer review reports and author responses from that submission.
Round 1
Reviewer 1 Report
Comments and Suggestions for Authors
Dear Authors, This is a good study and well written. Ethics approval was taken. It is problematic to analyse endometrial biopsy in patient with IUD.
I just need one clarification. Did you look at the types of IUD? did you look at any other hormonal treatment that patient might be taking?
was there any cases of finding actinomoycosis with IUD biopsies?
how do you explain the conclusion that thinner endometrium is associated with advanced endometrial pathology?
thank you
Author Response
Please find the reply in the attachment.

Reviewer 2 Report
Comments and Suggestions for Authors
Dear author
I read your paper with great interest about the impact of intrauterine devices on the quality of the endometrial samples among women with abnormal uterine bleeding. The topic is of global concern and the study has important clinical and management implications for women's health. Although the idea is not novel, the good sample size could shed more light on the outcome and clinical practice. I do praise the efforts made in preparing and conducting the study. The statistical tests and analysis were appropriate. However, several limitations may hinder the application of the results obtained and the study's validity.
1- The retrospective nature of the study and the old references (more than 66% of the references are older than 5 years) made the introduction somewhat outdated.
2- Not having the nature of the IUD in consideration, especially that you classified the histopathological results based on the hormonal effect. This would make the hormonal IUD a great bias if not taken into account.
3- You did not specify whether the women with AUB received any medical treatment before having the sampling process. This affects the pathological report.
4- You mentioned that women with insufficient samples were excluded from the study, and still, you grouped women with insufficient samples among category 3. "The third category consisted of unclassifiable cases, where specific hormonal influences were indeterminate, or insufficient biopsy material was obtained. This group included findings such as endometritis, inadequate samples, and signs of atrophy, often associated with inflammation, atrophy, or diagnostic challenges."
5- You did not justify why a thinner endometrium was associated with more advanced pathological changes. However, no significant association was found between IUD use and endometrial thickness, and the IUD users and non-users did not show a statistical difference.
6- In the discussion section, you mentioned that "For example, Critchley et al. (2020) reported that thinner endometrial structures are often associated with atrophy and malignancy [3]. Our findings support this hypothesis and indicate the need for careful histopathological evaluation in patients with thinner endometrial structures." How did you reach this conclusion, although you have excluded women with hyperplasia and malignancy from your enrollment criteria?
7- The clinical skills of the operator were not reflected in your study; this impacts the sampling volume presented to the pathology department for interpretation.
8- How can you explain that the ultrasound measurement of the endometrial thickness among IUD and non-IUD groups is almost the same, despite the separation of the two uterine walls by the IUD.
8- The first two paragraphs of the conclusion are a repetition of the introduction. These should be removed.
Comments on the Quality of English Language
The English language is excellent
Author Response

(The authors gave the same response as above.)

Reviewer 3 Report
Comments and Suggestions for Authors
- I read with great interest the Manuscript titled “The effect of intrauterine device on the quality of sampling material in patients undergoing endometrial biopsy" which falls within the aim of the Journal. In my honest opinion, the topic is interesting enough to attract the readers’ attention. Methodology is accurate and conclusions are supported by the data analysis. Nevertheless, authors should clarify some point and improve the discussion citing relevant and novel key articles about the topic. Authors should consider the following recommendations:
- Manuscript should be further revised by a native English speaker
- The manuscript does not distinguish between Cu++ and LNG-based IUDs. As these may induce different endometrial responses, stratification could have added to the findings to increase the generalizability. Please emphasize such limitation more adequately.
- Although the study highlights the limitations in biopsy quality due to IUD-induced changes, it fails to discuss or even mention office hysteroscopy as an alternative diagnostic and therapeutic tool. This is a missed opportunity, especially considering that outpatient hysteroscopy allows direct visualization and targeted biopsy with higher accuracy (see and cite: PMID: 37061093; PMID: 34752269) , which may circumvent the limitations of blind techniques in IUD users (see and cite: PMID: 35122582; PMID: 30138610)
- The exclusive use of Pipelle for blind sampling is a limitation. The authors should acknowledge and emphasize that this method may be not useful in most cases and contrast it with visual-guided techniques, as supported by recent guidelines, for most intrauterine pathologies in which visualizing the endometrium for macroscopic classification criteria is crucial (PMID: 40073921; PMID: 37811835; PMID: 34905882).
- Line 131–135: The classification system used for categorizing pathological findings based on hormonal influence is not standardized. Please clarify its clinical validation or provide a recent reference.
-
Abstract: Consider adding the retrospective nature of the study for transparency.
Author Response

(The authors gave the same response as above.)
